# Synthesis and Antiparasitic Activity of New Trithiolato-Bridged Dinuclear Ruthenium(II)-arene-carbohydrate Conjugates

**DOI:** 10.3390/molecules28020902

**Published:** 2023-01-16

**Authors:** Isabelle Holzer, Oksana Desiatkina, Nicoleta Anghel, Serena K. Johns, Ghalia Boubaker, Andrew Hemphill, Julien Furrer, Emilia Păunescu

**Affiliations:** 1Department of Chemistry, Biochemistry and Pharmaceutical Sciences, University of Bern, Freiestrasse 3, 3012 Bern, Switzerland; 2Institute of Parasitology Vetsuisse Faculty, University of Bern, Länggass-Strasse 122, 3012 Bern, Switzerland; 3School of Chemistry, Cardiff University, Park Place, Cardiff CF103AT, UK

**Keywords:** ruthenium(II)-arene complexes, bioorganometallic, carbohydrates, CuAAC reactions, antiparasitic, *Toxoplasma gondii*, human foreskin fibroblasts, auxotrophy, toxicity

## Abstract

Eight novel carbohydrate-tethered trithiolato dinuclear ruthenium(II)-arene complexes were synthesized using CuAAC ‘click’ (Cu(I)-catalyzed azide-alkyne cycloaddition) reactions, and there in vitro activity against transgenic *T. gondii* tachyzoites constitutively expressing β-galactosidase (*T. gondii* β-gal) and in non-infected human foreskin fibroblasts, HFF, was determined at 0.1 and 1 µM. When evaluated at 1 µM, seven diruthenium-carbohydrate conjugates strongly impaired parasite proliferation by >90%, while HFF viability was retained at 50% or more, and they were further subjected to the half-maximal inhibitory concentration (IC_50_) measurement on *T. gondii* β-gal. Results revealed that the biological activity of the hybrids was influenced both by the nature of the carbohydrate (glucose vs. galactose) appended on ruthenium complex and the type/length of the linker between the two units. **23** and **26**, two galactose-based diruthenium conjugates, exhibited low IC_50_ values and reduced effect on HFF viability when applied at 2.5 µM (**23**: IC_50_ = 0.032 µM/HFF viability 92% and **26**: IC_50_ = 0.153 µM/HFF viability 97%). Remarkably, compounds **23** and **26** performed significantly better than the corresponding carbohydrate non-modified diruthenium complexes, showing that this type of conjugates are a promising approach for obtaining new antiparasitic compounds with reduced toxicity.

## 1. Introduction

The interest in the development of metal complexes for medicinal applications increased in the middle of the 20th century after the discovery of the anticancer properties of cisplatin [1,2]. Metal-based drugs are attractive due to their great versatility in terms of metal center, oxidation state, coordination number, in addition to the nature and geometric orientation of the ligands [3]. As the use of platinum-based drugs is limited due to shortcomings like the occurrence of chemoresistance and side effects associated to their high toxicity [4,5], this encouraged the research of compounds based on other metals as alternative to platinum anticancer therapeutics [1,6,7,8]. Parallel investigations aimed to enlarge the purpose of metal complexes with the identification of additional pharmacological properties, such as antibiotic [9,10] and antiparasitic [11,12,13,14,15,16].

Ruthenium complexes were identified amid the most promising non-platinum chemotherapeutic alternatives [17,18]. The ruthenium(II)-arene scaffold has been declined in a myriad of compounds aimed to improve anticancer activity and selectivity [19,20,21,22,23,24,25], but also targeting other therapeutic applications [11,13,16,26].

A particular class of compounds containing this unit are the trithiolato-bridged dinuclear ruthenium(II)-arene complexes (**A**–**C** in Figure 1), which show not only high antiproliferative activity against cancer cells [27], but also promising antiparasitic properties [28,29]. The structure of these complexes is based on a trigonal bipyramidal Ru_2_S_3_ framework, with two ruthenium(II)-arene half-sandwich units. Two types of complexes can be distinguished, “mixed” (at least one of the bridge thiols is different, **A** in Figure 1) and “symmetric” (the three bridge thiols are identical, **B** and **C** in Figure 1) [27]. Former studies on *Toxoplasma gondii* [28], *Neospora caninum* [29] and *Trypanosoma brucei* [30] identified high antiparasitic activity for some of these diruthenium compounds. For example, compounds **A**–**C** (Figure 1) inhibit *T. gondii* tachyzoites proliferation with IC_50_ values in nanomolar range (down to 1.2 nM for **A**).

*T. gondii* is an obligate intracellular protozoan parasite of the phylum Apicomplexa that causes infections of medical and veterinary significance in humans and animals [31,32]. Infection is usually asymptomatic in immunocompetent individuals, but it may cause severe complications or even be fatal in immunocompromised patients [33]. Current common treatments for toxoplasmosis are not specific, require prolonged courses and have toxic side effects, and consequently, new therapeutic solutions are needed [33,34,35,36]. Unlike other pathogens, *T. gondii* has adapted to replicate in all nucleated cells of a wide range of vertebrates, regardless of their cellular metabolism, and thus displays an exceptional metabolic robustness [37,38]. *T. gondii* is auxotrophic for several metabolites including purines, polyamines, cholesterol and choline [37]. Accordingly, tackling the parasite auxotrophies and metabolic peculiarities can constitute an interesting therapeutic strategy [37].

Carbohydrates contribute to cell-cell recognition and adhesion, have a crucial role in cellular energy supply, and can bind to specific proteins (e.g., lectins, glucose transporters, and glycoenzymes). Consequently, their conjugation to metal complexes appears as a rational choice for drug design as it can promote biocompatibility and increase water solubility. Carbohydrate-metal hybrids show promise not only in medicinal chemistry [39,40,41] but also in catalysis [42,43]. Apart the metal, its oxidation state and coordination mode [44,45,46], various structural adjustments were considered as the type of the carbohydrate [47,48,49,50,51], its substitution position [52], and the presence and nature of the protecting groups [53,54,55]. The cancer cells glucose metabolism can be exploited for targeted therapy [56], and consequently, glycoconjugates of various metal complexes were explicitly designed for selective uptake by cells overexpressing glucose transporters [57,58,59]. In this context, the potential of ruthenium complexes containing carbohydrate-functionalized ligands has been extensively studied especially on cancer cells [60,61,62,63,64,65,66,67,68,69,70,71,72,73], and some representative examples are presented in Figure 1. Apart cancer-specific treatment [39,40,41,60,74,75,76,77], alternative utilizations of metal-carbohydrate hybrids, as for example antiparasitic therapy, also received a lot of interest [78,79,80] (Figure 1).

For example Ru(II)-arene complexes as **D** [65,66], with a carbohydrate-derived phosphorus-containing ligand, **E** [67] bearing a mannose fragment as a diamino-bidentate leg ligand, **F** [68,69], with a galactose fragment *N*-coordinated via a nitrile group, and **G** [71], containing a glucosyl functionalized 1,2,3-triazolylidene *N*-heterocyclic carbene ligand, exhibited promising antiproliferative activity on various cancer cells. Complexes like **H** [72], with methyl mannose or glucose units attached to a pyridyl-2-triazole bidentate ligand, were shown to exploit the glucose transporters for cellular uptake in cancer cells. For Ru(II) half-sandwich complexes like **I** [70] and **J** [73], the presence and nature of the protective groups proved to be essential for the biological activity. The high affinity of the malaria parasite for glucose was targeted using the ferrocenyl-glucose conjugate **K** [79], with moderate antimalarial activity in vitro in both *Plasmodium falciparum* chloroquine-resistant and non-resistant strains. Carbohydrate-ferrocenyltriazole conjugate **L** [81], exhibited antibacterial activity against both Gram-positive and Gram-negative pathogens, and triazole bridged ferrocene-selenoribose conjugate **M** [82] was cytotoxic on cancer cells.

This study continues the quest for trithiolato-bridged dinuclear ruthenium(II)-arene compounds as potential anti-*Toxoplasma* compounds with improved therapeutic value (in terms of antiparasitic efficacy/host cell toxicity balance) by exploiting the conjugate strategy and the parasite auxotrophies and specific metabolic needs. The investigation of carbohydrate metabolism in *T. gondii* has received a lot of interest [83,84,85,86] and considering the high energetic demand accompanying parasite growth and proliferation, carbohydrates can constitute an appealing choice among the metabolites able to promote the internalization of the organometallic unit in the parasite.

The synthesis of trithiolato diruthenium complexes is generally straightforward and efficient [87,88,89], this scaffold being robust to chemical modification and easily adaptable to the conjugate strategy as demonstrated by the various series of hybrids with peptides [90], drugs [91,92], fluorophores [89,93] or metabolites [93]. Ester and amide couplings [89,94], but also CuAAC (Cu(I)-catalyzed azide-alkyne cycloaddition) click reactions [92,93] proved to be useful tools for the functionalization of the diruthenium trithiolato unit at the level of the bridge thiols. CuAAC offer the advantage of mild reaction conditions, compatible with various ligands [44,46,71,72,73,95,96,97] but also with organometallics [81,98,99,100,101,102], and enables the construction of libraries of compounds [103,104,105]. Additionally, trithiolato diruthenium(II)-arene compounds suitably substituted with alkyne or azide groups were already used in CuAAC reactions for obtaining conjugates with molecules of interest e.g., various nucleic bases or drugs [92,93].

The nature of the carbohydrate (acetyl protected glucose or galactose) and the type and length of the linker between the two units were addressed as sources of variability. The new diruthenium hybrids and intermediates were screened in vitro against *T. gondii* tachyzoites expressing β-galactosidase (*T. gondii* β-gal) grown in human foreskin fibroblasts (HFF) with complementary assessment of HFF host cells viability. Compounds with promising antiparasitic activity and selectivity were then subjected to dose-response (IC_50_) determination on *T. gondii* β-gal and toxicity assessment on HFF at 2.5 M concentration.

## 2. Results and Discussions

### 2.1. Synthesis

#### 2.1.1. Synthesis of the Dinuclear Ruthenium(II)-arene Intermediates **2**–**9**

Alkyne and azide partners are needed for the CuAAC reactions, and when appropriately substituted, both the diruthenium moiety and the carbohydrate can play either role. With this aim, various diruthenium and carbohydrate intermediates were synthesized.

The dithiolato derivative **1** [106] (obtained from the ruthenium dimer ([(η^6^-p-MeC_6_H_4_Pr^i^)RuCl]_2_Cl_2_) and 4-*tert*-butylbenzenemethanethiol) was reacted with a second thiol (4-mercaptophenol, 4-aminobenzenthiol, 2-(4-mercaptophenyl)acetic acid, and 2-mercaptobenzyl alcohol, respectively) to provide the trithiolato-bridged dinuclear ruthenium compounds **2**–**5**, as previously reported (Figure 1) [87,88,89].

Intermediates **2**–**4** can be modified using ester and amide coupling reactions as previously described [89,92,93,94]. The alkyne ester **6** was obtained in moderate yield (47%) by reacting **2** with 5-hexynoic acid using EDCI (*N*-(3-dimethylaminopropyl)-*N′*-ethylcarbodiimide hydrochloride) as coupling agent, in basic conditions (DMAP, 4-(dimethylamino)-pyridine) (Figure 2, top). 5-Hexynoic acid was also reacted with the amino diruthenium derivative **3** using EDCI and HOBt (1-hydroxybenzotriazole) as coupling agents, in basic conditions (DIPEA, *N*,*N*-diisopropylethylamine), to afford the amido alkyne compound **7** as reported [93] (Figure 2, top). Similar reaction conditions were used for the synthesis of amide **8** from carboxylic acid diruthenium derivative **4** and propargylic amine as formerly described (Figure 2, bottom) [92,93].

The azide trithiolato diruthenium derivative **9** (Figure 3), was obtained following a two steps pathway starting from alcohol **5** using a reported protocol [93]. First, the hydroxy group was activated by mesylation (MsCl, methanesulfonyl chloride) in basic conditions (TEA, triethylamine), followed by the nucleophilic substitution with azide (NaN_3_).

#### 2.1.2. Synthesis of the Azide and Alkyne Functionalized Carbohydrate Intermediates **10**–**18**

Appropriate carbohydrate derivatives bearing azide and alkyne groups were also synthesized (Figure 4 and Figure 5). Azido glucose compound **10** (2,3,4,6-tetra-O-acetyl-β-D-glucopyranosyl azide), was synthesized from commercially available 2,3,4,6-tetra-O-acetyl-β-D-glucopyranosyl bromide following a literature protocol (Figure 4) [107]. The reaction was realized with TMS-N_3_ (trimethylsilylazide) in THF in the presence of TBAF (tetrabutylammonium fluoride) in catalytic amounts, and **10** was isolated in moderate yield (51%).

The azide compounds **14**–**16** were obtained following a two-step procedure previously described [52,55,108] (Figure 5, top). First, *β*-D-glucose pentaacetate and *β*-D-galactose pentaacetate were glycosylated with 2-bromoethanol and 4-bromo-1-butanol. The reactions were realized in the presence of BF_3_∙Et_2_O (boron trifluoride diethyl etherate) as Lewis acid catalyst [52] and afforded the ether glycosides **11**–**13** in low to moderate yields (32, 41, and 47%, respectively). In the second step the bromine atom on the pending chain of **11**–**13** was substituted with azide (NaN_3_) [55,108], derivatives **14**–**16** being isolated in 47, 74% and quantitative yields, respectively.

The alkyne functionalized carbohydrates **17** and **18** were synthesized (Figure 5, bottom) from *β*-D-glucose pentaacetate and, respectively, from *β*-D-galactose pentaacetate and 4-pentyn-1-ol in the presence of BF_3_∙Et_2_O [109] and were isolated in medium yields (64 and 46%).

#### 2.1.3. Synthesis of the Carbohydrate Functionalized Trithiolato-Bridged Dinuclear Ruthenium(II)-arene Complexes **19**–**26**

The carbohydrate units were attached to the trithiolato diruthenium scaffold via click 1,3-dipolar cycloadditions using adapted protocols [110,111,112], in the presence of CuSO_4_ as catalyst and sodium ascorbate as a reducing agent, in DMF under inert conditions. Complexes **6**–**9**, bearing either alkyne or azide pendant group, were reacted with the appropriately functionalized carbohydrate derivatives **10** and **14**–**18** (Figure 6, Figure 7, Figure 8 and Figure 9) affording eight new trithiolato diruthenium conjugates **19**–**26**.

Thus, alkyne functionalized diruthenium compounds **6** and **7** were reacted with glucose derivative **10** presenting an azide group directly anchored to the glucopyranosyl ring (Figure 6). Amide conjugate **20** was isolated in good yield (72%), while difficulties were encountered in the purification of ester analogue **19** which was recovered in poorer yield (28%).

Other glycoconjugates were synthesized using alkyne intermediate **7** (Figure 7) and two types of modifications were envisioned: (i) the nature of the carbohydrate (glucose in **21** vs. galactose in **22**), and (ii) the presence of spacers of different length between the azide group and the glucopyranosyl ring (galactose derivatives **22** and **23**). Conjugates **21**–**23** were isolated in good yields of 66, 65 and 74%, respectively. Neither the steric hindrance nor the nature of the carbon atom on which the azide group was anchored (**10** vs. **14**) play a key role on the yield (**20** vs. **21**–**23**). Similarly, the reaction of the diruthenium propargyl amide derivative **8** with galactose azide **14** afforded conjugate **24** in 75% yield (Figure 8).

The trithiolato dinuclear intermediate **9**, presenting an azide in benzylic position on one of the bridge thiols, was reacted with glucose and galactose alkyne derivatives **17** and **18** (Figure 9), affording carbohydrate conjugates **25** and **26** isolated in moderate yields of 51 and 63%, respectively.

All compounds were fully characterized by ^1^H, ^13^C nuclear magnetic resonance (NMR) spectroscopy, high resolution electrospray ionization mass spectrometry (HR ESI-MS) and elemental analysis (see Appendix A). The obtainment of the triazole connector between the diruthenium unit and the carbohydrates was undoubtedly demonstrated by the ^1^H and ^13^C NMR spectra of the conjugates 19–26 by the signals corresponding to the proton of the triazole cycle at 7.74–8.66 ppm and of the corresponding carbon at 120.3–124.6 ppm. The absence of the signals corresponding to the proton of the monosubstituted alkyne (at 1.93–2.49 in compounds 6–8, 17 and 18) in the ^1^H NMR spectra of conjugates 19–26 further confirm the obtainment of the hybrid molecules. Mass spectrometry corroborated the spectroscopic data with the trithiolato diruthenium glucose and galactose conjugates **19**–**26** showing molecular ion peaks corresponding to [M-Cl]^+^ ions.

#### 2.1.4. Stability of the Compounds

For the assessment of the biological activity, the compounds were prepared as stock solutions in dimethylsulfoxide (DMSO). Similar to former reports [88,89,94], the ^1^H NMR spectra of the functionalized diruthenium complexes **6**, **7**, **20**, **22**, **25** and **26** in DMSO-*d_6_*, recorded at 25 °C 5 min and more than 1 month after sample preparation showed no significant modifications (see Appendix A), demonstrating a very good stability of the compounds in this highly complexing solvent.

Compound **19** has an ester linker that can potentially be hydrolyzed in cell growth media. Comparable conjugates with fluorophores (coumarin and BODIPY) linked through ester bonds to the trithiolato diruthenium unit were recently studied [89,93]. Only very limited solvolysis of the ester bonds was noticed after 168 h for some compounds, and it was concluded that the fluorophore diruthenium conjugates exhibit high stability in the conditions used for the biological evaluations. Therefore, it was assumed that compound **19** is appropriately stable for the first in vitro biological activity evaluation.

### 2.2. Assessment of the In Vitro Activity against T. Gondii β-gal and Human Foreskin Fibroblast Host Cells

#### 2.2.1. Primary Screening

The biological activity of the carbohydrate azides and alkyne derivatives **14**–**16** and, respectively, **17** and **18** was not measured as these compounds were not isolated pure. Glucose and galactose conjugates **19**–**26**, glucose azide derivative **10** and diruthenium alkyne intermediate **6** were assessed for their in vitro biological activity in inhibiting proliferation of *T. gondii* β-gal, a transgenic strain that constitutively expresses β-galactosidase, and for toxicity to HFF (human foreskin fibroblast) used as host cells. The compounds were applied to infected or non-infected HFF cultures for 72 h and at concentrations of 0.1 and 1 µM, the results being summarized in Table 1 and Figure 2. The viability of treated HFF was measured by the alamarBlue metabolic assay, and the proliferation of *T. gondii* β-gal was quantified by the β-galactosidase colorimetric test. In both cases, results are expressed as percentage (%) compared to control parasitic and host cells treated with 0.1% DMSO for which proliferation and viability were set to 100% (Table 1).

The trithiolato diruthenium complexes **2**–**5** and **9**, and alkyne intermediates **7** and **8** were evaluated previously against *T. gondii* β-gal under similar conditions [88,89,92,93], and the corresponding values are shown in Table 1 and Figure 2 for comparison. The new alkyne ester derivative **6** impacted *T. gondii* proliferation but affected significantly less the viability of HFF compared to its diruthenium hydroxy intermediates **2**. The glucose azide derivative **10** exhibited neither antiparasitic activity nor host cell toxicity at both tested concentrations.

In the first screening, the eight carbohydrate conjugates **19**–**26,** applied at 1 µM, did not impair host cell viability. Apart from glucose conjugate **25**, all the dyads nearly abolished parasite proliferation when applied at 1 µM. Though, apart from glucose conjugate **19**, all hybrid molecules had only a limited effect on *T. gondii* β-gal at 0.1 µM.

Both the type of carbohydrate and the nature and length of the linker influenced the antiparasitic efficacy and cytotoxicity of the conjugates, but no clear straightforward trends could be identified regarding the relationship between the structural elements and biological activity. For example, when the conjugates are applied at 0.1 µM, some differences in anti-*Toxoplasma* efficacy are observed. For instance, glucose ester derivative **19** is significantly more active on *T. gondii* compared to the amide analogue **20**. Galactose functionalized compound **22** is more efficient in inhibiting the parasite proliferation compared to the corresponding glucose derivative **21**, while for the same carbohydrate an increase of the linker length has a negative effect on the antiparasitic activity (galactose conjugates **22** and **23**).

#### 2.2.2. IC_50_ Values against *T. gondii* β-gal Tachyzoites and HFF Toxicity at 2.5 µM

For a compound to be selected for the second screening, two criteria had to be met simultaneously: (i) when the compound was applied at 1 µM, *T. gondii* β-gal growth was inhibited by 90% or more compared to control treated with 0.1% DMSO only, and (ii) HFF host cell viability was not impaired by more than 50% for a compound applied at 1 µM. Based on the results of the primary screening, glucose and galactose dyads **19**–**24** and **26** were selected. Pyrimethamine, currently used for the treatment of toxoplasmosis, and which inhibited the proliferation of *T. gondii* β-gal tachyzoites with an IC_50_ value of 0.326 µM and did not affect HFF viability at 2.5 µM (Table 2), was used as reference compound. The selection also included the diruthenium intermediate compounds **2**, **3** and **5** with free OH or NH_2_ groups, along with two diruthenium alkyne ester and amide compounds **6** and **7**, and diruthenium azide **9**. The results are summarized in Table 2.

The IC_50_ values and the cytotoxicity of the diruthenium compounds **2**, **3**, **5**, **7** and **9** were measured previously [88,89,92,93]. For these diruthenium intermediates the IC_50_ values ranged from 0.025 µM (**6**) to 0.153 µM (**3**). However, all intermediates also strongly affected the viability of HFF when applied at 2.5 µM, the most cytotoxic being compounds **5**, **6** and **9**.

Glucose conjugates **19** and **21** exhibited low IC_50_ values (0.018 and 0.087 µM, respectively), but were toxic to host cells at 2.5 µM (HFF viability was reduced to 29% for **19** and abolished for **21**). Glucose hybrid **20** exhibited antiparasitic activity (IC_50_ = 0.110 µM) but also medium cytotoxicity (HFF viability of 77%). Galactose and glucose dyads **22** and **24** had only modest antiparasitic activity (IC_50_ values of 0.294 and 0.328 µM, respectively, comparable with those obtained for pyrimethamine) while being moderately toxic to HFF at 2.5 µM (73 and 66%, significantly more cytotoxic compared to the standard pyrimethamine).

Galactose conjugates **23** and **26** were the most promising of the series exhibiting not only high efficacy in inhibiting *T. gondii* β-gal proliferation (IC_50_ values of 0.032 and 0.153 µM, 10-fold and 2-fold lower compared to pyrimethamine, IC_50_ = 0.326 µM), but also low cytotoxicity on the host cells when applied at 2.5 µM (HFF viability 92 and 97%, respectively).

Interestingly, both glucose and galactose hybrids **20** and **23** affected the HFF viability less than the diruthenium alkyne intermediate **7** from which they were obtained by click reactions. A similar result was also obtained for the galactose conjugate **26** compared to the diruthenium azide parent **9**.

The number of conjugates considered in this study is too limited to allow proper SAR observations. Nevertheless, apart from the conjugation with protected carbohydrates, other structural features of the dyads (as the nature and length of the linker between the two units), appear to strongly influence the biological activity, and a fine structural tuning is needed to obtain compounds with good pharmacological properties in terms of safety/anti-toxoplasma efficacy balance.

Further studies are necessary for the identification of the mode of action of trithiolato diruthenium compounds. For some other types of dinuclear Ru(II)-arene complexes reported in the literature, interactions with DNA and oligonucleotide sequences were identified [113,114,115,116,117,118]. However, unlike other Ru(II)-arene complexes presenting labile chlorine, carboxylate or monodentate N-coordinated ligands, the trithiolato diruthenium complexes do not hydrolyze and are stable in the presence of most biomolecules such as amino acids and DNA [27]. Furthermore, a recent study revealed only weak interactions via H-bonding nucleobase-pairing between trithiolato diruthenium nucleobase conjugates and the respective complementary nucleic bases [93]. In the presence of some trithiolato diruthenium complexes the oxidation of cysteine (Cys) and glutathione (GSH) to form cystine and GSSG, respectively, was observed [119,120]. TEM (transmission electron microscopy) studies of different protozoan parasites (*Toxoplasma gondii*, *Neospora caninum*, *Trypanosoma brucei*) treated with trithiolato dinuclear ruthenium(II)-arene complexes revealed alterations in the mitochondrial ultrastructure indicating this parasite organelle as potential target [29]. Noteworthy, trithiolato diruthenium conjugates with coumarin and BODIPY fluorophores [89,121] induced analogous outcome on parasite mitochondrion.

## 3. Materials and Methods

### 3.1. Chemistry

The chemistry experimental part, with full description of synthetic procedures and characterization data for all compounds are presented in the Appendix A.

### 3.2. Biological Evaluation

#### 3.2.1. Cell and Parasite Culture

All tissue culture media were purchased from Gibco-BRL, and biochemical agents from Sigma-Aldrich. Human foreskin fibroblasts (HFF) were obtained from the American Type Culture Collection (ATCC) and maintained in complete culture medium consisting in DMEM (Dulbecco’s Modified Eagle’s Medium) supplemented with 10% fetal calf serum (FCS, Gibco-BRL, Waltham, MA, USA) and antibiotics as previously described [122]. Transgenic *T. gondii* β-gal tachyzoites (expressing the β-galactosidase gene from Escherichia coli) from RH strain were kindly provided by Prof. David Sibley (Washington University, St. Louis, MO, USA) and were maintained by passages in HFF cultures as previously described [122,123].

#### 3.2.2. In Vitro Activity Assessment against *T. Gondii* Tachyzoites and Human Foreskin Fibroblasts

The screening sequence for the compounds was described in previous reports [88]. All compounds were prepared as 1 mM stock solutions from powder in dimethyl sulfoxide (DMSO, Sigma, St. Louis, MO, USA). For in vitro activity and cytotoxicity assays, HFF were seeded at 5 × 10^3^/well in 96 well plates and allowed to grow to confluence in complete culture medium at 37 °C and 5% CO_2_. Transgenic *T. gondii* β-gal tachyzoites were freshly isolated from infected cultures as described [122], and 96-well plates containing HFF monolayer were infected with 1 × 10^3^ tachyzoites/well.

In a primary screening, each compound was evaluated at two concentrations 0.1 and 1 µM and added to the media prior to the infection as previously described [94]. Control non-infected non-treated HFF cultures and *T. gondii* β-gal infected but not-treated cultures were cultivated in complete medium containing 0.01 or 0.1% DMSO. The 96-well plates were incubated for 72 h at 37 °C/5% CO_2_ as previously described [94].

For the IC_50_ determination on *T. gondii* β-gal, eight serial concentrations ranging from 7 nM to 1 μM were tested for each selected compound as previously described [89,92,93] and the β-galactosidase assay was performed as reported [122]. Briefly, infected HFF cultures in 96-well plates were lysed with PBS containing 0.05% Triton X-100. Then the substrate chlorophenolred-β-D-galactopyranoside (CPRG; Roche Diagnostics, Rotkreuz, Switzerland) was added in a final concentration of 0.5 mM. Absorption was measured at 570 nm wavelength using an EnSpire^®^ multimode plate reader (PerkinElmer, Inc., Waltham, MA, USA).

All calculations were performed using the corresponding software tool contained in the Excel software package (Microsoft, Redmond, WA, USA). Cytotoxicity assays using uninfected confluent HFF host cells were performed by the alamarBlue assay as previously reported [124]. Confluent HFF monolayers in 96-well plates were exposed to 0.1, 1 and 2.5 μM of each compound and incubated for 72 h at 37 °C/5% CO_2_. Then the medium was removed, the plates were washed once with PBS and 200 μL of resazurin (1:200 dilution in PBS) were added to each well. Plates were measured at excitation wavelength 530 nm and emission wavelength 590 nM using an EnSpire^®^ multimode plate reader (PerkinElmer, Inc.). Fluorescence was measured at two different time points: T_0_ as starting timepoint and T_5h_ as at 5 h later. Relative fluorescence units were calculated from time points with linear increases.

## 4. Conclusions

This study was focused on the synthesis and in vitro anti-*Toxoplasma* activity evaluation of eight new trithiolato-bridged arene-ruthenium(II) carbohydrate conjugates. Acetyl protected glucose and galactose moieties were pended on the diruthenium unit on one of the bridging thiols using CuAAC click reactions and connectors of several types and lengths to obtain the carbohydrate dyads. In the first screening, none of the conjugates affected the validity of host cells at 1 µM, suggesting reduced toxicity, and seven carbohydrate-diruthenium hybrids applied at 1 µM inhibited *T. gondii* β-gal growth by more than 90%. The second screening (IC_50_ values and toxicity to HFF after exposure to 2.5 µM) led to the identification of two promising acetyl protected galactose functionalized compounds **23** and **26**. Both conjugates not only exceeded (up to 10-fold) the anti-*Toxoplasma* efficacy of the standard drug pyrimethamine for similar level of toxicity to HFF, but also exhibited a significantly better antiparasitic activity/cytotoxicity balance compared to the corresponding carbohydrate non-modified diruthenium complexes.

The type and length of the linker between the diruthenium core and the carbohydrate unit significantly influenced the biological activity, and fine structural adjustments could further increase the anti-*Toxoplasma* efficacy of this type of carbohydrate conjugates. In addition, the nature of the carbohydrate and the presence/absence of protecting groups is known to strongly affect the biological activity of conjugates carbohydrate-organometallic complex [53,70,73]. Thus, the use of other carbohydrates bearing, or no protective groups is also considered.

This study showed that carbohydrate conjugation to trithiolato-diruthenium complexes is a promising strategy for obtaining novel organometallic compounds with high antiparasitic efficacy and reduced host cell cytotoxicity.

## Data Availability

The data is included in the article and the Appendix A.

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
