# Peer review of "Synthesis and Antiparasitic Activity of New Trithiolato-Bridged Dinuclear Ruthenium(II)-arene-carbohydrate Conjugates"

_molecules, 2023, doi:10.3390/molecules28020902_

Round 1

Reviewer 1 Report

The present article “Synthesis and Antiparasitic Activity of New Trithiolato-Bridged Dinuclear Ruthenium(II)-Arene-Carbohydrate Conjugates” describes the synthesis, characterization, and biological activity against Toxoplasma gondii tachyzoites of eight new carbohydrate-tethered trithiolato dinuclear ruthenium(II) complexes. The syntheses through CuAAC ‘click method are well presented and the compounds are accurately structurally characterized by classical methods (NMR, ESI-MS, and EA). The methodology used in the biological assays it is also well-described and reproducible. The results of the biological tests are relevant relating to the conclusions. A promising generation of compounds with antiparasitic activity has been prepared and tested on T. gondii and HFF non-infected fibroblasts. The manuscript respects the aims and scope of the special issue “Metal-Based Drugs: Past, Present and Future“ and therefore I find it suitable for publication in Molecules with minor revisions. Although, I have two recommendations for the authors:

1.      Taking into consideration that metal complexes have been synthesized herein, the most accurate structural determination would be by X-Ray diffraction. In case of a poor crystallization of these types of complexes, and no possibility of diffraction measurements, I would suggest to the authors to add in the structural characterization also the  Infra-Red spectra measurements for complexes.

2.      My second comment is concerning the information given in Subchapter 2.1.2 lines 291-293 about the relationship between the type of carbohydrate and/or the nature of the linker which according to the authors, does not influence the antiparasitic activity. In Conclusions Chapter, lines 412-414, it is summarized the importance of the linkers in the anti-Toxoplasma efficacy. These two sentences induce a kind of confusion, therefore I would recommend a reformulation of the information in the text, either by avoiding one of the sentences or by correlating them accordingly to the content.

Author Response

Reviewer 1

The present article “Synthesis and Antiparasitic Activity of New Trithiolato-Bridged Dinuclear Ruthenium(II)-Arene-Carbohydrate Conjugates” describes the synthesis, characterization, and biological activity against Toxoplasma gondii tachyzoites of eight new carbohydrate-tethered trithiolato dinuclear ruthenium(II) complexes. The syntheses through CuAAC ‘click method are well presented and the compounds are accurately structurally characterized by classical methods (NMR, ESI-MS, and EA). The methodology used in the biological assays it is also well-described and reproducible. The results of the biological tests are relevant relating to the conclusions. A promising generation of compounds with antiparasitic activity has been prepared and tested on T. gondii and HFF non-infected fibroblasts. The manuscript respects the aims and scope of the special issue “Metal-Based Drugs: Past, Present and Future“ and therefore I find it suitable for publication in Molecules with minor revisions.

  • We thank this reviewer for his general positive comments.

Although, I have two recommendations for the authors:

  1. Taking into consideration that metal complexes have been synthesized herein, the most accurate structural determination would be by X-Ray diffraction. In case of a poor crystallization of these types of complexes, and no possibility of diffraction measurements, I would suggest to the authors to add in the structural characterization also the Infra-Red spectra measurements for complexes.

  • It’s a matter of fact that we could not grow suitable crystals for those carbohydrate conjugates. We measured IR spectra about 12 years ago on our first Diruthenium prototypes, but we quickly realized that IR spectra were, for non-symmetric compounds (like those presented in this work) mostly, excessively complicated and uninformative. We therefore no longer characterize our Diruthenium compounds and conjugates by IR spectroscopy. Indeed, IR data may precious indications on the obtainment of the conjugates via CuAAC coupling reactions following for example the IR absorbance band of the azide (N3) group in compounds 9, 10, 14-16. However, the NMR data (Supporting information) unquestionably confirm the obtainment of the triazole bonded conjugates and an additional comment was introducing in the text of the manuscript (lines 230-236 Page 8). The obtainment of the click products was also proved by ESI-MS measurements (Supporting information).

  1. My second comment is concerning the information given in Subchapter 2.1.2 lines 291-293 about the relationship between the type of carbohydrate and/or the nature of the linker which according to the authors, does not influence the antiparasitic activity. In Conclusions Chapter, lines 412-414, it is summarized the importance of the linkers in the anti-Toxoplasma efficacy. These two sentences induce a kind of confusion, therefore I would recommend a reformulation of the information in the text, either by avoiding one of the sentences or by correlating them accordingly to the content.

  • We thank the reviewer for this important comment.

For clarity and for avoiding confusion, we have modified the sentence (lines 297-300, page 11).

Reviewer 2 Report

1. Authors showed that carbohydrate conjugation to trithiolato-diruthenium complexes is a promising strategy for obtaining novel organometallic compounds with high antiparasitic efficacy and reduced host cell cytotoxicity using very basic assays.  Previous literature revealed that Dinuclear Ruthenium(ll)-Arene have potential to interact with DNA and induce apoptosis. At least, it will be better if you provide morphological assessments.

2.   Authors have discussed about very basic about diruthenium and the carbohydrate complex influenced the biological activity. But, they did not discussed the mode of action.

Author Response

  1. Authors showed that carbohydrate conjugation to trithiolato-diruthenium complexes is a promising strategy for obtaining novel organometallic compounds with high antiparasitic efficacy and reduced host cell cytotoxicity using very basic assays.  Previous literature revealed that Dinuclear Ruthenium(ll)-Arene have potential to interact with DNA and induce apoptosis. At least, it will be better if you provide morphological assessments.

  1. Authors have discussed about very basic about diruthenium and the carbohydrate complex influenced the biological activity. But, they did not discussed the mode of action.

The authors thank the reviewer for the suggestions and certainly this type of experiments would be interesting to be realized in future studies. Further investigations allowing to clarify the mechanisms that lead to the antiparasitic activity are aimed for some of the compounds of this study. Comments regarding the possible mechanism of action of diruthenium trithiolato compounds were introduced in the text of the manuscript Lines 357-372, Page 13-14. The scope of this study was mainly to identify if targeting T. gondii metabolic pathways using carbohydrate conjugates of trithiolato diruthenium derivatives can constitute a promising approach. The strategy was validated as some of the hybrid molecules presenting protected carbohydrate appendices performed better compared to parent complexes lacking this type of substituents.